# Driver Liability Assessment in Vehicle Collisions in Spain

**DOI:** 10.3390/ijerph18041475

**Published:** 2021-02-04

**Authors:** Almudena Sanjurjo-de-No, Blanca Arenas-Ramírez, José Mira, Francisco Aparicio-Izquierdo

**Affiliations:** 1Instituto Universitario de Investigación del Automóvil Francisco Aparicio Izquierdo (INSIA-UPM), Escuela Técnica Superior de Ingenieros Industriales (ETSII), Universidad Politécnica de Madrid (UPM), 28006 Madrid, Spain; blancavarenas@gmail.com (B.A.-R.); francisco.aparicio@upm.es (F.A.-I.); 2Statistics Department, Escuela Técnica Superior de Ingenieros Industriales (ETSII), Universidad Politécnica de Madrid (UPM), 28006 Madrid, Spain; josemanuel.mira@upm.es

**Keywords:** road safety, vehicle collisions, pattern identification, driver liability assignment, Self-Organizing Maps (SOM), quasi-induced exposure

## Abstract

An accurate estimation of exposure is essential for road collision rate estimation, which is key when evaluating the impact of road safety measures. The quasi-induced exposure method was developed to estimate relative exposure for different driver groups based on its main hypothesis: the not-at-fault drivers involved in two-vehicle collisions are taken as a random sample of driver populations. Liability assignment is thus crucial in this method to identify not-at-fault drivers, but often no liability labels are given in collision records, so unsupervised analysis tools are required. To date, most researchers consider only driver and speed offences in liability assignment, but an open question is if more information could be added. To this end, in this paper, the visual clustering technique of self-organizing maps (SOM) has been applied to better understand the multivariate structure in the data, to find out the most important variables for driver liability, analyzing their influence, and to identify relevant liability patterns. The results show that alcohol/drug use could be influential on liability and further analysis is required for disability and sudden illness. More information has been used, given that a larger proportion of the data was considered. SOM thus appears as a promising tool for liability assessment.

## 1. Introduction

In road safety research, a critical point is the estimation of vehicle collision risks and rates of the different driver groups (male vs. female, drivers in different age groups and so forth), with the aim of establishing preventive measures which try to avoid crashes or, at least, to minimize their impact, as pointed out by many researchers such as [1,2,3,4,5,6,7,8,9,10,11,12,13].

However, to assess the risk levels of the different driver groups, it is necessary to count on some measure of the exposure levels of the driver groups or vehicles which are being analyzed, given that vehicle collision rates are defined as the ratio between the number of collisions in the group and their exposure [1,4,14]. The problem here lies in the exposure term, which is not available in most databases and its determination is not an easy task, particularly in the more specific (restricted) groups of drivers or under several risk combinations [2,15,16], so this constitutes one of the most important issues for vehicle collision analyses. This occurs especially in countries like Spain, where the surveys for determining the exposure of the different driver groups are not carried out routinely [1,9,12,14].

The quasi-induced exposure method is selected here to estimate the relative exposure of the drivers studied and also the crash-causing propensity by means the relative accident involvement ratio (RAIR), which is calculated by taking the ratio of the number of at-fault drivers in a specific subgroup to that of the not-at-fault drivers in the same subgroup [15,17,18]. The quasi-induced exposure method uses the vehicle collision data on the basis of several hypotheses [1,4,8,9,10,12,13,17,19,20].

The theory of the quasi-induced method presents two basic underlying hypotheses: (a) data from exclusively clean collisions—where only one of the drivers has been classified as at-fault—are to be applied, and (b) the not-at-fault drivers in clean two-vehicle collisions constitute a sample of the total drivers in specific time and place as highlighted by [1,2,3,5,6,7,8,9,10,11,13,14,18,20,21,22,23].

Therefore, one of the main issues when estimating the relative exposure lies in the correct liability assignment of the collision based on the information provided in the vehicle collision records. This very relevant issue has been debated since the beginning of the history of the quasi-induced techniques [21]. The Spanish General Road Crashes database does not include specific information about the liability of each driver who is involved in a crash, although it includes the offences committed by drivers and their conditions [11,12]; for example: speed offences or alcohol/drugs use.

It is observed that many authors, like [3], have applied the “contributing human factor” for determining collision liability, and this considers both hazardous driving behavior (e.g., not to observe a Stop signal or distracted driving) and the non-driving behaviors (e.g., driving whilst under the influence of drugs or alcohol) [7,8]. In turn, other authors, such as [13,24] took the “driver’s citation” (any citation by the police) as a starting point to determine driver liability in collisions [5,6,7,8]. The remaining scientific literature, mostly, use different combinations of the two above mentioned factors, like [10,25] or do not specify in detail which criteria to use to determine vehicle collision liability. Presumably, the driver’s state citation could be used as a warrant for liability assignment [4,8,9,23]. A more thorough review may be found in [7,8].

Therefore, in most cases, the not-at-fault driver is defined as the one who does not provide any contributing human factor for the occurrence of the vehicle collision and has not received any citation by the police [1,3]. However, adding information on non-driving behavior or on the driver’s state citations could result in uncertainty or statistical bias in exposure estimation [5,6,7,8,20,26]. For example, there exists in police citations, the risk of “negative halo bias” [26]. An example of this phenomenon was illustrated by [7], who discovered that young male drivers using alcohol/drugs were more likely to be receive a police citation, which would in turn bias citation sampling [6,7,8].

This is why, mainly during the last few years, researchers have mostly been inclined to consider essentially hazardous driving behavior (mainly driver and speed offences) to assign driver liability in the collision, considering that, in the literature to date, the quasi-induced exposure method is based on driver behavior and condition [6,7,8]. However, there are driver impairments or offences regarding driver behavior and condition, for example alcohol or drugs use, which increase the probability that a driver is at fault, especially if they are combined with other variables. Nevertheless, they are usually not considered because they are not per-se determinant in the liability assignment so they could, in a deterministic approach, bias the process. This approach is considered a priori too restrictive by the authors of this paper, who believe that there are additional variables which individually or jointly could affect driver liability, so this issue should be explored. Therefore, incorporating additional variables regarding driver behavior or condition in liability assignment, applied to the quasi-induced exposure method, could be considered.

Given that there are no liability labels in the vehicle collision records, it is not possible to apply supervised analysis techniques such as logistic regression or multiple regression models, to estimate driver liability in terms of driver offences and condition variables. Therefore, an unsupervised analysis, which takes into account these variables, is called for. The authors believe that patterns can be identified, in terms of driver offences and conditions during collisions, which would correspond to categories in the degree of certainty about the driver liability: very clear, clear, likely, unclear, and so forth. These patterns would correspond to relatively homogenous clusters in terms of the collision variables above. Moreover, given the potentially complex multivariate structure of the data, there are patterns which would only come to light when many offence and condition data are analyzed together. Thus, a cluster analysis is called for as a useful tool to support and improve the quality of liability assignment.

Therefore, the main contributions of this paper are: (1) to explore the inclusion of more variables in the liability assignment procedure applied to the quasi-induced exposure method, given that in the literature, mainly in recent years, only driver and speed offences have been taken into account; and (2) to apply a powerful clustering (thus unsupervised analysis) technique such as self-organizing maps (SOM), for expert judgment-based liability assignment in terms of the patterns identified from the clustering. Using this clustering and expert judgment, liability regions are identified in terms of (sometimes complex) multivariate patterns. In addition, as an added value, the SOM also provides a better understanding of the multivariate structure of the data. The use of an unsupervised analysis technique, such as SOM, implies an important methodological contribution to liability assignment, given that to date there has existed no systematic statistical methodology to this end, and that there are potentially relevant variables which were not taken into account in the literature and could also affect liability. Thus, this paper contributes to improving the quality of the liability assignment applied to the quasi-induced exposure method. The latter, as pointed out above, is key when evaluating the impact of road safety measures because it is used to estimate relative exposure, which is essential for road collision rate estimation, and thus it has successfully been applied to policy-making.

Therefore, SOM has been applied to a database on collisions between two passenger cars in order to: (a) cluster the data in terms of the eight offence variables to observe how they are grouped in accordance with their multivariate structure; (b) select the variables which are relevant in the definition of the clusters and which should thus be considered when identifying patterns; and (c) based on (a) and (b), undertake pattern identification directed to liability assignment given that, as mentioned above, there are no liability labels attached to the data and the analysis is unsupervised. In addition, the results will be compared to driver liability assignment based only on driver and speed offences, which are the factors most commonly used by researchers to make the assignment.

## 2. Materials and Methods

### 2.1. Database

To carry out this research, the corresponding vehicle collision database was obtained from the General Road Crashes database provided by the Spanish Traffic General Directorate (Dirección General de Tráfico, DGT). The final database only included vehicle collisions in interurban areas between two passenger cars that occurred in Spain from 2004 through 2013. At first, this involved a total of 836,598 drivers, whose information was provided (gender, age, offences, injury severity, etc.), as well as on their vehicle (vehicle defect, color, etc.) and the collision characteristics (type, location, day of the week, etc.). Each record in the base had data on a single driver, i.e., there were two records per collision, one for each driver. Thus, the number of drivers analyzed was equal to the number of records and twice the number of vehicle collisions.

There were three reasons for the choice of passenger cars. First, according to the data from the Spanish Traffic General Directorate, in the study period of this paper (2004–2013), passenger cars account for more than 70% of vehicles. Secondly, the number of victims in interurban road crashes is also above 70% of the grand total. Third, passenger cars are the vehicle group where quasi-induced exposure methods have been more frequently applied. It is thus a very important group for road safety research.

In addition, interurban areas were chosen because the number of killed and seriously injured drivers in interurban areas is much larger than in urban ones. In particular, in the 2004–2013 period analyzed here, the figures for killed and seriously injured for interurban areas were 4 times and twice those of the urban ones, respectively.

The period 2004–2013 was chosen because, from 2014 on, the road crashes data have been registered by means of the online application ARENA (Accidents: information collection and analysis). Thus, there are two road crashes databases: the first is the General Road Crashes database, which includes the vehicle collisions from 1993 to 2013, and the second is the ARENA database, which includes the vehicle collisions from 2014 onwards and so far, the two databases have had a different format and structure, mainly for fatalities. Work is currently being carried out so that, in future research, information on the recent years may be added, thus increasing the size and hence also the accuracy of the statistical inference. However, a large sample size is used for the research in this paper.

The screening and data cleaning process of the database (Figure 1) was carried out in two stages: (I) in the beginning, the baseline data was screened to keep only two vehicle crashes (head-on, off-set frontal, side and rear-end collisions) and subsequently restricted to interurban areas (main roads and alternative ways) and two passenger cars, so that the database was reduced to a total of 146,162 drivers; (II) next, it was necessary to carry out a data cleaning given that errors in the data were still detected and there was a lack of information on some drivers, etc. This process was carried out with the R program, which is a language and environment for statistical computing and graphics [27], and with which the final database was reduced to a total of 145,904 records of drivers involved in road crashes.

In this research, all driver behavior- and condition-related variables, which, to a greater or less degree, have any influence over the driver liability, have been considered, given that this research is a contribution to improve liability assignment applied to the quasi-induced exposure method, which is in turn driver behavior-based, as pointed out by other authors [6,7,8,19]. As mentioned above, the “traditional” procedures of the quasi-induced exposure method usually only take into account driver and speed offences, but here 8 variables were used instead, i.e., more information. In Table 1 the potentially relevant variables and their most common or representative types, are shown.

The driver and speed offences should be included in the analysis because they are related to hazardous driving behavior and, as such, it was expected that they were the best to determine driver liability.

It is expected that the rest of the variables introduced in the model provide additional information about driver liability because, although they alone are believed to be non-determinant in the assignment of liability, it is possible that the interactions between several of them will be influential because the joint effect of subset of variables could be more relevant in liability than their individual effects. Therefore, it is interesting to evaluate their joint influence on all drivers.

A priori, data on basic demographics, such as gender or age, could have also been taken into account. However, none of the corresponding variables have been mentioned in the literature as relevant, or even potentially relevant, to liability assignment in the quasi-induced exposure method. In addition, other variables related to the collision characteristics or vehicle could be also considered. However, the quasi-induced exposure method is based on the driver behavior and conditions. Therefore, only driver behavior- and condition-related variables are considered.

In addition, as some researchers pointed out [6,19,28], the original databases have usually some problems such as incomplete or non-valid information, unknown values, under-reporting problems mainly in no-injury or less severe-injuries collisions, etc., it is thus important to use the maximum amount of information possible to analyze vehicle collisions. Therefore, in this work, all the variables that have unknown values in the records have been taken into account.

### 2.2. Methodology

The methodology applied through this research is cluster analysis, namely SOM. The self-organizing map was developed by Kohonen (1990) and is one of the most popular neural networks and a well-known technique for clustering and visualization, which can be included within the machine learning techniques. It belongs to the competitive learning networks category, where the different nodes (clusters) in the map compete for the data assignment [29,30].

The purpose of SOM is to represent and cluster multidimensional data in a space of smaller dimension, typically 2D, so that the clustering can be visualized in a so-called map, while maintaining the topological structure, i.e., in such a way that sample points which are close in the original space will still be so in the reduced one. This dimensionality reduction feature is a great advantage of SOM because, as mentioned above, the 2D map will provide a visible and thus very rapidly analyzable clustering [31,32,33,34].

The SOM algorithm is fundamentally developed in four stages: initialization, competition, cooperation and adaptation [33,35]. It proceeds sequentially, i.e., each sample point is assigned in turn to the nearest node in the map, and subsequently both the weights of the winner node and the neighboring ones are updated. This update of the neighboring nodes is called the cooperative learning stage. It is what provides the aforementioned preservation of topology.

The final result of SOM is thus a map of a finite number of nodes, indexed by pairs of integer numbers, where every sample point of the original (high dimensional) space is assigned to a single node of the 2D map. The nodes in the map also have a representation in the original high dimensional space, which are named their weights.

A certain limitation of SOM is that, since it performs multicriteria optimization (on both cluster homogeneity and conservation of topology), then if one just wanted to optimize on the first criterion, it (SOM) would be suboptimal and one should use traditional (non-projected) cluster techniques (e.g., K-means) instead.

Moreover, the number of variables in a SOM is limited by its main advantage with respect to other clustering techniques: visibility (and hence quickness of understanding of the clustering structure). One may thus pay a price for this advantage in terms of information. More information regarding the SOM methodology can be found in [33,35].

The choice of SOM is justified because the absence of liability labels in the data calls for the use of unsupervised learning techniques, such as cluster analysis. Within clustering tools, SOM has, with respect to other clustering techniques, the advantage of visibility. Therefore, it is a powerful visualization tool for data analysis which provides a better understanding of the multivariate data, whose joint dependence structure is taken into account in the liability assignment process. Thus, with the SOM methodology, driver behavior pattern identification regarding driver impairments or offences can be carried out. The joint analysis of all the offence variables can shed light on driver liability by assigning expert judgment-assessed liability labels to the SOM clusters.

## 3. Categorization of the Variables: Sensitivity Analysis of Self-Organizing Maps (SOM) with Sample Selection

To be able to work with the variables in Table 1 applying the SOM methodology, it has been necessary to transform the categorical variables into numerical ones. In this section, the process of variable categorization will be described.

The 0 value has been used to indicate that this offence or unfavorable condition is not present, while the 2 value has been used if it is. Given that all variables used in the SOM have been transformed from their original categorical values to discrete values between 0 and 2, their range is irrelevant or, in other words, their ranges are “standardized”.

The problem of coding is mainly found in the cases in which the values of one or more of the variables analyzed are unknown. There has been a tendency to believe that if the value of a given variable is unknown, the value to be assigned to it should be the average of those associated to lack (0) and to the presence (2) of these offences or impairments, that it to say, in our case it will have to take the value 1. However, it is considered here that the problem is more complex and, to choose this value the starting point is the two following hypotheses: (a) it should lie between 0 and 2, given that it is an intermediate category, and (b) if there is an “unknown” value or a datum which is not registered by the police, it is more likely that this could be due to no offence, so it should be assigned a value closer to 0 than to 2, but it is difficult to establish a specific one. Thus, to assess this question from the point of view of its effect on the SOM, in this section a SOM-based sensitivity analysis using a sample of the database is carried out and compared with three different values: (A) the value 0.25 is taken to represent the unknown values of the variables; (B) likewise for 0.5; and (C) likewise for 1. To obtain these maps, the SOM’s seed has been held fixed so that the random initialization of the algorithm affects equally all the maps.

Two SOMs are exactly the same if the relative positions of the data are the same for both maps [36,37]. Therefore, if two SOMs are equal, the distance between pairs of drivers located in the clusters of the first SOM should be equal to the respective distances between the nodes which contain that same couple of drivers in the second SOM [37].

Since 3 different maps have been obtained (for 0.25, 0.5 and 1), pairwise comparisons have been carried out. For each of these maps, the distance matrices between the sample’s drivers have been calculated. However, since comparing distance matrices for the full database of 145,904 drivers is unfeasible, a subsample (M) of size n has been drawn and distance matrices of size n × n have been compared. This matrix has zeros in all the elements of the diagonal and is symmetric, given that, in a same SOM map, the distance between a driver and him/her self is zero and the one between drivers A and B is the same as the one between B and A. Thus, a distance matrix is obtained between the n drivers for each one of the SOM maps, which were compared as set out in Figure 2.

To choose the sample size of n drivers for distance matrix comparisons, it should be taken into account that there are, as far as the authors now, no theoretical results on sample representativeness for similarity between SOMs, i.e., results on the sampling distributions of these similarity measures between distance matrices.

However, there do exist analytical results for estimates of proportions which could provide an insightful reference [38]. For a 0.05 maximum estimation error for the proportion *p* under random sampling, since the width of the corresponding confidence interval is:(1)$$1.96*\sqrt{\frac{p\left(1-p\right)}{n}},$$
where n is the sample size, if one chooses the most unfavorable case to be conservative, that would be *p* = 0.5, that is:(2)$$1.96*\sqrt{\frac{0.5*\left(1-0.5\right)}{n}}=0.05;\;n=384$$

Therefore, a sample size of 384 drivers is taken as a reasonable value for distance matrix comparisons.

When carrying out these distance matrix comparisons in R, the results which are obtained are those shown in Figure 3.

A priori, it is uncertain if these differences can be significant or not. They should thus be standardized or compared with any type of internal “reference” error of the algorithm. Namely, the error or variability due to initialization has been chosen as reference.

To this end, a total of eight SOMs with all the drivers of the database have been obtained with different initializations but holding a value of 0.25 fixed for the cases in which the value of a variable is unknown. It is assumed that the variability due to initialization will be similar for 0.5 and 1. As outlined above, a sample of size n = 384 is taken again for the distance matrix comparisons. This would provide potentially 28 pairwise distance matrix comparisons, but only 14 randomly selected comparisons have been actually carried out. The results of 4 of these 14 comparisons are shown in Figure 4.

The average percentage difference of all comparisons, i.e., the internal variability of SOM due exclusively to different initialization, is approximately 16%. This value is only slightly higher than the aforementioned variability due to using 0.25, 0.5 or 1, i.e., the latter is not really significant.

Given that the choice of 0.25, 0.5 or 1 is not that significant, the analysis has been carried out taking 0.25 which is considered nonetheless by expert opinion as the most suitable for these situations.

Thus, the final variables and their categorization are shown in Table 2.

## 4. Results and Discussion

This section has been divided in four subsections. In the first one, the offences and conditions SOM is shown. In the second one, it is assessed which are the variables that define the SOM clusters. In the third subsection, pattern identification is carried out in terms of these variables, in order to help in the liability assignment process. Finally, in the last subsection, the results with SOM are compared with those of the traditional liability assignment process.

### 4.1. SOM Division of the Multivariate Structure

In this subsection it is explained how the drivers are allotted along the SOM map so that the drivers’ data with similar multivariate characteristics are included in the same node or in close ones of the map.

The distribution of the 145,904 drivers along the offences SOM is shown in Figure 5.

In the map, drivers are distributed in 25 clusters and the driver topology characteristics can be visualized in the 2D space when, originally, it was eight-dimensional (one for each offence variable). The proportion of drivers in each cluster is shown below (within each cluster). The cluster numbers are indicated in red. There exist quantitative criteria for selecting the number of clusters. For example, in probabilistic mixture model-based clustering, the EM (Expectation Maximization) algorithm may be applied. Here, the choice of 25 nodes was made empirically by trial and error: SOMs with different map sizes have been obtained, starting with the smallest ones. 25 nodes were considered a reasonable choice, given the trade-off sought between properly identifying patterns (clarity/visibility) and sample size per cluster: with an excessively small map size, clusters could be too heterogeneous and, therefore, adequate patterns would not be extracted; the same would occur with a very large map size, resulting in too small cluster sample sizes [39]. Also, 25 clusters is an adequate number to provide a better understanding of the multivariate structure of the data.

In this map, the average values of each variable in each cell (the circular sectors within a cluster) are shown with different colors. This representation helps to visualize the joint (multivariate) structure in the original data space. Each circular sector radius within a cluster will be smaller or larger (the angle is the same for all variables, 360/number of variables) depending on the average value (over all the drivers in the cluster) of the variable it represents. The radius will be maximum when either (a) the value of the variable in question for all drivers is 2, which means that all the drivers in the cluster have this offence or impairment or (b) the average of the variable in this cluster is larger than any other one (cluster), whereas it will be minimal when the average is 0 (the circular sector is not represented for that variable) and, therefore, no driver in the cluster will have committed that offence or present the impairment that the variable indicates [39].

The weights of the different clusters are shown in Table 3. Clusters 10, 13, 17 and 23 have not been taken into account, given that there are zero drivers assigned to them, they are thus transition clusters, necessary to preserve/reflect distance and for this reason, they are maintained in the map, but will not affect the rest of analysis.

Table 3 is important because, although the weights are already illustrated in Figure 5, their exact values are given therein. Therefore, the SOM and its corresponding weight table have to be analyzed together.

In accordance with the information presented in Figure 5 and Table 3, there are two important SOM clusters, number 5 and number 15, which include more than 50% of the total driver sample. Cluster 5 is formed by all drivers who committed no offences and presented no unfavorable conditions for driving, this is thus why it includes so many drivers. On the other hand, cluster 15 includes all drivers who have committed some driver offence. There are 22 types of such offences and, additionally, some of them are relatively frequent (e.g., distractions). Thus, this cluster also contains many drivers.

### 4.2. Most Influential Variables on Driver Liability

SOM will be now used to provide an auxiliary tool to determine which variables are more relevant to the liability assignment applied to the quasi-induced exposure method. This will be done by selecting the variables which play a significant role in defining the SOM clusters, while also leaning on prior knowledge on vehicle collision liability.

This issue is important given that, as mentioned above, it has been observed that, so far, the procedure for such assignment to date mainly takes into account only driver and speed offences and, in addition, it is not clear which additional explanatory variables regarding driver behavior or condition should or should not be introduced in this process.

In any research on the liability assignment in terms of the variables of the vehicle collision database, driver and speed offence have to be taken into account. In addition, it is reasonable to assume that the more clearly a variable divides the SOM, the more relevant to driver liability it will be. In Figure 5 and Table 3, it can be observed that the variable which best divides the SOM is driver offence, given that, within all clusters, it only takes the values 0 or 2 (Table 3) and it is the one that clearly divides the map into two halves. Therefore, it would potentially be the variable which best defines driver liability.

In addition, in the SOM the “Disability” and “Vehicle defect” variables appear isolated in a few clusters (1 and 2, respectively), i.e., with negligible mean values for the remaining ones. Therefore, these variables should perhaps not be taken into account in the liability assignment process. However, disability is mainly related with the age of the drivers [39] and elderly drivers are more likely than younger ones to be at-fault in a crash [15]. Thus it could be useful to include this variable in order to obtain more information about driver behavior patterns. More studies should be carried out to evaluate the disability variable influence over the driver liability.

As for administrative offences, the only one which could be more relevant for liability assignment is not passing the roadworthiness test (MOT) when it appears jointly with a vehicle defect. However, no clusters have been found in which both offences coexist. This is why this variable is considered not very important in liability assignment.

A similar situation is found with “Drowsiness”, which is usually absent. In addition, in the SOM no node has been identified in which only that situation appears, given that it happens together with driver offence, whose weight to determine liability is clearer. This suggests that this variable should not be considered.

As for the “Sudden illness” variable, the type of sudden illness suffered by drivers is not specified in the collision database, but if available would facilitate its use. It could include those conditions that appear without being expected and usually cause loss of the normal condition of drivers, for instance: fainting, brain injury, heart attack, anxiety and so on. Therefore, they can affect the likelihood of being or not at-fault in the collision, given their influence on driver ability. However, in general, there are only a few drivers who suffer sudden illness and are usually spread out in the SOM and their influence in the liability assignment is thus usually low. Therefore it is not clear if this variable should be taken into account in the liability assignment process. The use of intelligent devices, which provide information about the driver state, would be useful in order to evaluate properly the relevance of sudden illness on the liability assignment.

The variable types classified by their level of influence on the liability assignment using SOM, are shown in Table 4.

### 4.3. Pattern Identification for the Liability Assignment Process

In this subsection, by using the results of the previous steps of SOM clustering and relevant cluster variable selection, pattern identification in terms of the selected variables is carried out in order to help with the liability assignment process.

In the SOM there are two areas which are clearly identified subjectively: liable (at-fault driver) and Presumed non-liable (not-at-fault driver) (Figure 6-the less relevant variables are also maintained). This boundary was established, given that SOM is an unsupervised technique, by expert criteria, based on the characteristics of the offences which drivers have committed or not in both map subsets i.e., the boundary in the map has been estimated depending on the “subset of offences committed by drivers” pattern in each cluster.

At the top of the map (68,730 at-fault drivers), the labeling errors should be minimal, given their profile of offences because this area includes all drivers who have committed offences which, in accordance to the literature reviewed, carry very large weights when establishing liability, e.g., speed offences. For this reason, the top of the map is named “Liable”.

The bottom of the map (77,174 drivers) is composed of all drivers who have committed an offence or have an impairment which increases the chance that a driver was at-fault, but does not per-se determine it. For this reason, the bottom of the map is named “Presumed non-liable”.

The 5th cluster, at the bottom of the map (presumed non-liable) should be highlighted. It consists entirely of not-at-fault drivers (45,006 drivers). However, in the remaining nodes in this area, as mentioned above, there is less clarity in driver liability, given that many less important variables in the liability assignment coexist and there are more variables whose values are unknown.

Therefore, to carry out driver classification for all nodes of the map, a joint analysis about drivers in these clusters and the data on their opposing driver in the collision was necessary. This analysis was carried out with the R software and the following cases are considered: (1) if both drivers at the same collision were assigned to any cluster at the top of the map, it would be a collision where both drivers can be considered as at-fault (7626 drivers); (2) if one of the two drivers involved in a collision is in the lower zone of the map (presumed non-liable), but the other one belongs to a cluster at the top of the map (liable), then the former could be considered not-at-fault or vice versa. There are 122,208 drivers in this category (61,104 at-fault drivers and 61,104 not-at-fault drivers); (3) if both drivers were in the 5th cluster (2014 drivers), then both should be labeled as not-at-fault; (4) if both drivers were assigned to two different clusters in the presumed non-liable zone, additional analyses about these drivers should be carried out to assess if it is possible to classify them or not, even though, in general, their classification is not possible with just the available data. The analysis for (4) has not been carried out in this research, therefore 14,056 drivers who meet this case criterion were rejected in the study.

In Table 5, the casuistry of classifying drivers, which has been detailed in the analysis above, is summarized. In it can be seen that, of the total drivers evaluated, 83.76% of them could be classified as part of a clean collision.

In Table 5, “Fault / Fault” and “Not-at-fault/Not-at-fault” categories are not taken into account because they are not clean collisions, an essential requirement in the quasi-induced exposure method. Therefore, only the drivers which are involved in clean collisions are considered (122,208 drivers).

### 4.4. Comparison of Results with the Traditional Liability Assignment Process

To support the validity of the results obtained, the liability assignment has been also carried out for comparison (reference) based only on driver and speed offences, which are the most commonly used by researchers to make the assignment, given that, as pointed above, they measure hazardous driving behavior. Thus, a driver is considered at-fault if he/she has committed a driver or speed offence and not-at-fault if that is not the case.

With this second criterion, a characterization in useful cases of 72.47% of the records is achieved for the quasi-induced exposure, as shown in Table 6. This is 11.29% below the 83.76% achieved with the new procedure of this paper.

In liability assignment based exclusively on driver and speed offences, the number of “Not-at-fault/Not-at-fault” cases will be significantly increased because the procedure considers that any drivers who have not committed such offences are not at fault. Therefore, cases in which, for instance, one of the drivers has a disability and the other driver has alcohol/drug use will be classified as not-at-fault/not-at-fault, while with the SOM methodology these drivers are in the “Other drivers to analyze” category.

In addition, when liability is based only on driver and speed offences, the number of cases of the “Other drivers to analyze” category is also increased, given what occurs in liability assignment through SOMs. This is because the assignment process based on these two variables does not take into account the cases in which some data are unknown. In other words, with the SOM methodology many cases in the “Other drivers to analyze” category (uncertain) are now assigned to fault/not-at-fault.

To summarize, the application of SOM to driver data provides identification, based on their offences, of patterns that can help in the liability assessment process.

## 5. Conclusions

The main contributions of this research are the following:-A better understanding of the complex multivariate structure of the data.-To provide a tool (SOM) to help identify driver liability patterns applied to the quasi-induced exposure method based on the database information.-Improvement of the quantity and quality of the data for future application of the quasi-induced exposure method, which is necessary to estimate the relative exposure level of the different driver groups. This is an important contribution to road safety research.

Given that no liability labels exist in many databases, the analysis is based on an unsupervised machine learning tool. This research has proposed an alternative methodology (self-organizing maps, SOM) to carry out a joint analysis of the variables which are considered relevant in assignment of liability in order to identify patterns which help to determine the most and the less influential variables on the driver liability and help to carry out the liability assignment itself in collisions between two passenger cars in interurban areas. This methodology has been applied given that it has been observed that the procedure for liability assignment mainly takes into account only driver and speed offences and a discussion has existed about which other variables best determine liability. Thus, the potential of this methodology was to take advantage of the opportunity of obtaining additional information based on more variables, by means of a better understanding of the multivariate structure of the data. Multivariate analysis is considered a more thorough solution than univariate or bivariate, given that some complex behaviors may only unfold when all variables are analyzed jointly. In addition, with this research, additional information about the characteristics of drivers involved in vehicle collisions as well as of the collisions themselves is extracted.

The analysis suggests that the most important variables for driver liability are: driver offence, speed offence and alcohol/drug use, although the two first ones are the most important. The relevance of the variables disability and sudden illness regarding the driver liability is not clear and administrative offence, drowsiness and vehicle defect are clearly the less-relevant ones, in such way that, when carrying out the liability assignment, not taking them into account would not be influential, thus simplifying the process (with respect to considering the eight initial offence variables). However, the patterns and thus the boundaries identified by SOM are more complex than those which would result from just taking into account driver and speed offences and alcohol/drug use. Some of the remaining variables may not be very relevant marginally, but could be so jointly with others. Thus, although the three first variables are the most relevant, some information is still contained in the two following ones.

Finally, liability assignment carried out based on the patterns identified through the SOM, by assigning, from expert opinion a label to each cluster, allows for potentially classifying 83.76% of the drivers, which means a higher proportion of driver assignments to at-fault or not-at fault status than with the standard procedure (72.47%). The quality of this classification was improved given that, with the methodology proposed in this paper, complex multivariate structure is taken into account.

The main limitations of the study are: (a) it is very important to choose properly the SOM variables because their number is limited by its visibility (because it is a projected clustering), which is, at the same time, the main advantage of this clustering technique with respect to other ones. SOM thus provides a very quick assimilation/description of the cluster structure of the data. One may thus pay a (small) price for this advantage in terms of information; (b) quasi-induced exposure is a method based on driver behavior and condition. Therefore, to assign driver liability, traditionally only driver behavior- and condition-related variables have been used. However, there are other variables related to vehicle collision circumstances, such as weather or pavement conditions, which could also affect, to a lesser degree, driver liability. Therefore, this issue should be explored in future research; and (c) as mentioned in the paper, more studies should be carried out with disability and sudden illness variables in order to evaluate their relevance on driver liability in depth.

From the point of view of the added value of this paper, this research includes important quantitative results. The paper thus goes clearly far beyond a purely binary or qualitative statement on whether given variables are influential or not on liability. This quantification includes the numbers (proportions) of drivers in each cluster/node as well as the SOM topology and neighbor structure. A complete description of the patterns identified necessarily includes this information. Moreover, the borders between clear, non-clear and intermediate liability assignments are also only fully specified if quantification is given.

Therefore, with the SOM methodology an improvement of the quality of estimated relative exposure based on quasi-induced exposure is expected. This is the result of taking into account more variables and data (also variables whose values are unknown have been considered). It will thus be possible to better understand a phenomenon as complex as vehicle collisions, as well as to establish different driving-related measures to reduce collisions, minimize their impact and improve road safety.

## Figures and Tables

**Figure 1 ijerph-18-01475-f001:**
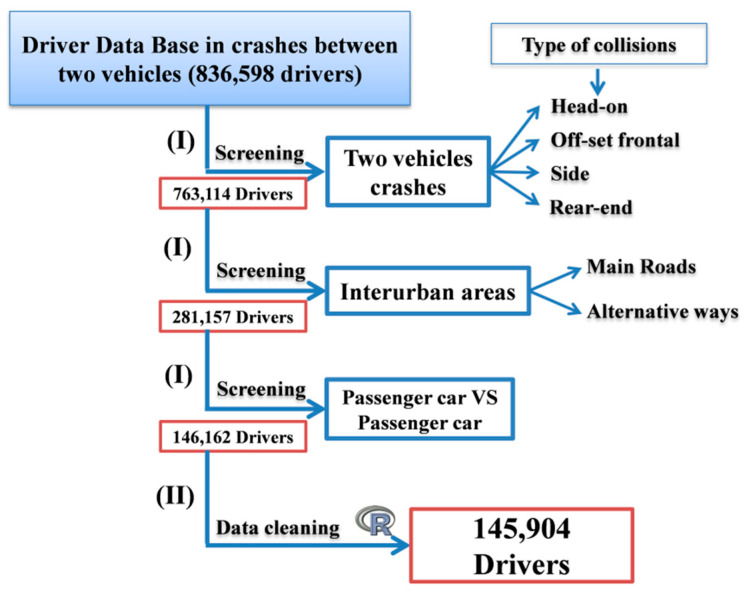
Screening and data cleaning process.

**Figure 2 ijerph-18-01475-f002:**
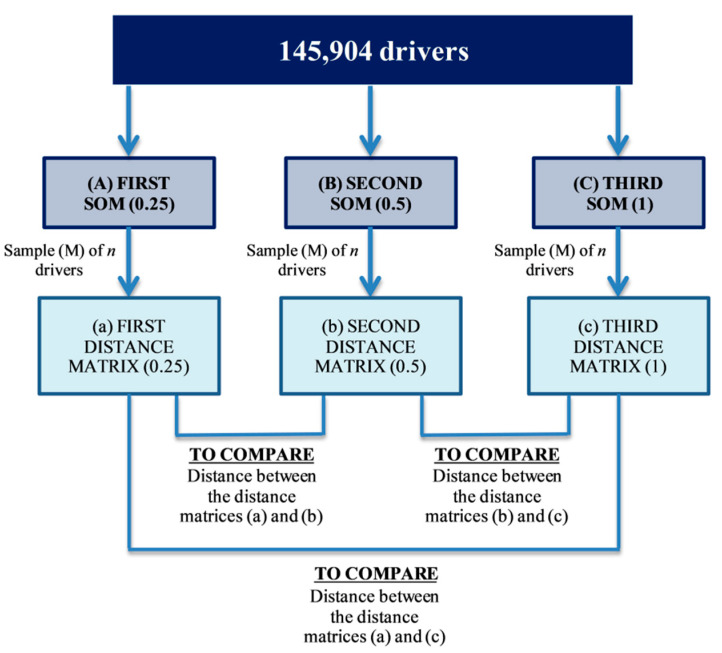
Pairwise comparison between self-organizing maps (SOMs) for 0.25, 0.5 and 1.

**Figure 3 ijerph-18-01475-f003:**
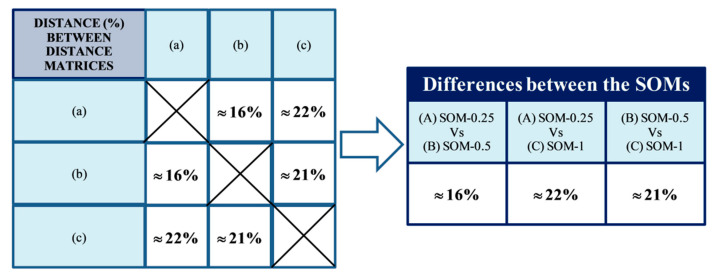
Percentual differences between the different distance matrices.

**Figure 4 ijerph-18-01475-f004:**
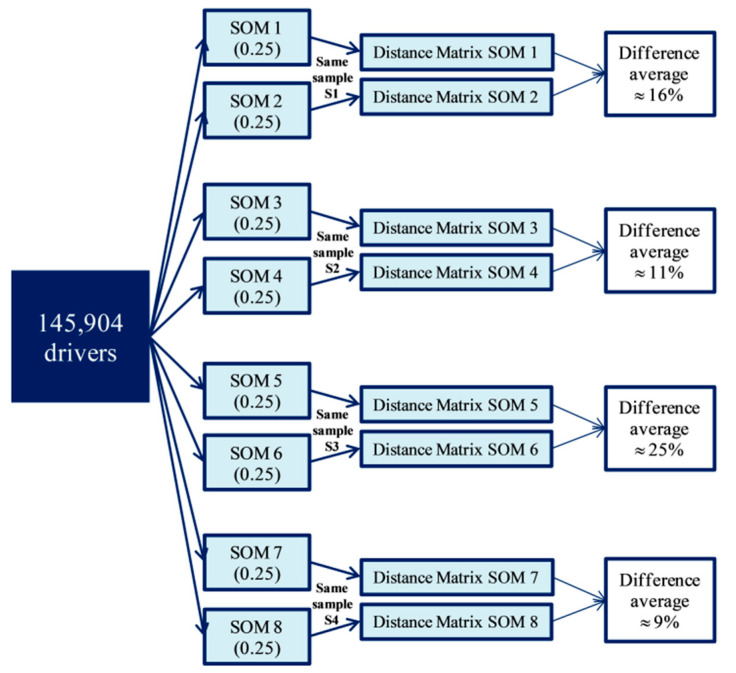
Internal SOM error estimation procedure.

**Figure 5 ijerph-18-01475-f005:**
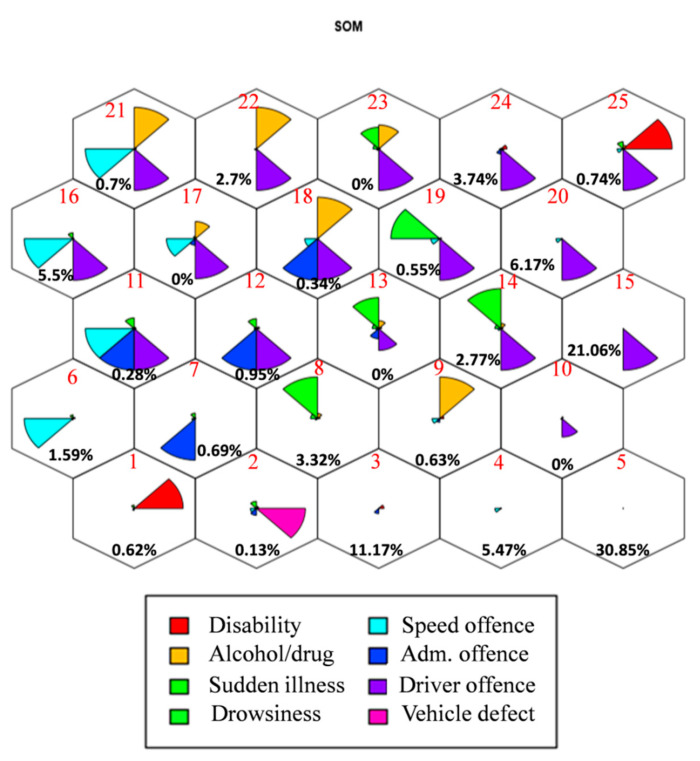
Self-organizing map for the driver offences.

**Figure 6 ijerph-18-01475-f006:**
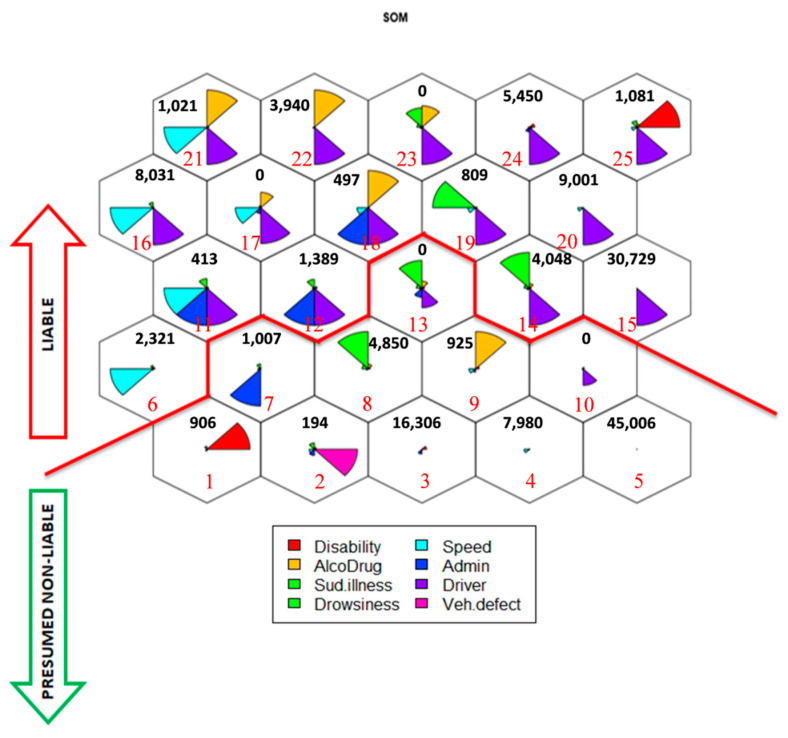
Self-organizing map with two different areas.

**Table 1 ijerph-18-01475-t001:** Potentially relevant (driver behavior and condition) variables.

Potentially Relevant Variables	Most Representative Types (Description)
Driver offence	Distracted driving Non compliance of the Stop signal Partial invasion of opposing lane Non compliance of the headway distance
Speed offence	Inadequate speed under existent conditions Driving above the speed limits Too slow driving disturbing traffic
Administrative offence	Invalid driver license Expired driver license Not passing the roadworthiness test (MOT: in Spain, it is a test which, by law, must be made periodically on all road vehicles that are more than 4 years old, in order to check that they are safe to drive)
Disability	Sight Hearing Lower–Upper limbs
Vehicle defect	Very worn out tyres Flat tyre Missing tyre Deficient front of rear—lights Deficient brakes
Alcohol/drugs use	These variables indicate: not respecting the limits of alcohol/drug during driving
Drowsiness	This variable indicates if the driver has or not drowsiness, fatigue or concern and it has been named “Drowsiness”.
Sudden illness	Sudden illnesses may be defined as those which appeared unexpectedly and usually cause loss of standard abilities. Examples of sudden illnesses: Passing out, epileptic seizure, heart attack, anxiety attack, etc.

**Table 2 ijerph-18-01475-t002:** Final variables and their categorization.

Variables	Categories	Values
Driver offence	No driver offence	0
	Driver offence	2
Speed offence	No speed offence	0
	Unknown	0.25
	Speed offence	2
Administrative offence	No administrative offence	0
	Unknown	0.25
	Administrative offence	2
Disability	No disability	0
	Unknown	0.25
	Disability	2
Vehicle defect	No vehicle defect	0
	Unknown	0.25
	Vehicle defect	2
Psychophysical Circumstances		
Alcohol/Drug use	No alcohol/drug use	0
	Unknown	0.25
	Alcohol/drug use	2
Sudden illness	No sudden illness	0
	Unknown	0.25
	Sudden illness	2
Drowsiness	No drowsiness	0
	Unknown	0.25
	Drowsiness	2

**Table 3 ijerph-18-01475-t003:** Average value (weight) of each variable in the different clusters.

	Number of Drivers	Disability	Alcohol or Drug	Sudden Illness	Drowsiness	Speed Offence	Administr. Offence	Driver Offence	Vehicle Defect
Cluster 1	906 (0.62%)	2	0.02	0.02	0.02	0	0.09	0	0
Cluster 2	194 (0.13%)	0.10	0.10	0.05	0.06	0.25	0.33	0	2
Cluster 3	16,306 (11.17%)	0.22	0	0	0	0	0.25	0	0
Cluster 4	7980 (5.47%)	0	0	0	0	0.25	0.03	0	0
Cluster 5	45,006 (30.85%)	0	0	0	0	0	0	0	0
Cluster 6	2321 (1.59%)	0.09	0.02	0.02	0.02	2	0.07	0	0
Cluster 7	1007 (0.69%)	0.02	0.03	0.03	0.07	0	2	0	0
Cluster 8	4850 (3.32%)	0.10	0.24	0.27	0.29	0	0.03	0	0
Cluster 9	925 (0.63%)	0.17	2	0	0	0.32	0.18	0	0
Cluster 11	413 (0.28%)	0.07	0.07	0.07	0.08	2	2	2	0.07
Cluster 12	1389 (0.95%)	0.13	0.05	0.06	0.08	0	2	2	0.05
Cluster 14	4048 (2.77%)	0.09	0.25	0.26	0.25	0	0.04	2	0
Cluster 15	30,729 (21.06%)	0	0	0	0	0	0	2	0
Cluster 16	8031 (5.5%)	0.02	0.04	0.04	0.04	2	0.03	2	0.03
Cluster 18	497 (0.34%)	0.05	2	0	0	0.52	2	2	0.04
Cluster 19	809 (0.55%)	0.07	0	0	2	0.36	0.02	2	0.02
Cluster 20	9001 (6.17%)	0	0	0	0	0.25	0.02	2	0
Cluster 21	1021 (0.7%)	0.09	2	0	0	2	0.03	2	0.02
Cluster 22	3940 (2.7%)	0.03	2	0	0	0	0.04	2	0.03
Cluster 24	5450 (3.74%)	0.25	0	0	0	0	0.24	2	0.12
Cluster 25	1081 (0.74%)	2	0.17	0.04	0.02	0.25	0.02	2	0

**Table 4 ijerph-18-01475-t004:** Categories for variable influence on the liability assignment process.

Most Influential Variables in the Assignment of Liability	Less Influential Variables in the Assignment of Liability	Unknown Relevance in the Liability Assignment Process
Driver offence	Administrative offence	Disability
Speed offence	Drowsiness	Sudden illness
Alcohol/drugs use	Vehicle defect	

**Table 5 ijerph-18-01475-t005:** Driver classification by SOM methodology.

Liability Categories	Number of Drivers	%
Fault/Not-at-fault	122,208	83.76%
Fault/Fault	7626	5.23%
Not-at-fault/Not-at-fault	2014	1.38%
Other drivers to analyze	14,056	9.63%
TOTAL	145,904	100%

**Table 6 ijerph-18-01475-t006:** Driver classification by driver and speed offences.

Liability Categories	Number of Drivers	%
Fault/Not-at-fault	105,736	72.47%
Fault/Fault	7706	5.28%
Not-at-fault/Not-at-fault	13,098	8.98%
Other drivers to analyze	19,364	13.27%
TOTAL	145,904	100%

## Data Availability

The data presented in this study are available on request from the corresponding author. The data are not publicly available due to the data were provided by the Spanish Traffic General Directorate (DGT) to the INSIA-UPM to be used exclusively for research purposes.

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
