# Peer review of "Driver Liability Assessment in Vehicle Collisions in Spain"

_ijerph, 2021, doi:10.3390/ijerph18041475_

Round 1
Reviewer 1 Report
The authors addressed all my concerns. Good job!
Author Response
Thank you very much for your comment. We are pleased to have addressed all your concerns.
Reviewer 2 Report
Thank you for our efforts on the paper. From a very established and long history of road safety research we already know that propensity to violate leads to higher crash risk, we already know that impairment via alcohol and drugs lead to greater crash risk we also know that age, gender and inexperience, socioeconomic status, are related to crash liability. This literature has not been referred to in this work. The work does not provide new insights in to crash liability.
Author Response
Many thanks for your comment. We are very disappointed to hear that you believe our paper does not provide new insights into crash liability. This paper aims to significantly contribute to improving the liability assignment applied to the quasi-induced exposure method. In the literature, mainly in recent years, only driver and speed offences are taken into account. In our paper, all the 8 variables, related to driver behavior and condition at the time of the vehicle collision, have been included (vs. 2 variables considered previously in the literature). The results obtained show how variables such as alcohol and / or drugs influence the liability of drivers, although it is usually not considered in driver liability assignment process. From the point of view of the added value of our paper, we would like to strongly emphasize that our analysis includes important quantitative results. The paper thus goes clearly far beyond a purely binary or qualitative statement on whether given variables are influential or not on liability. This quantification includes the numbers (proportions) of drivers in each cluster/node as well as the SOM topology and neighbor structure. A complete description of the patterns identified necessarily includes this information. Moreover, the borders between clear, non-clear and intermediate liability assignments are also only fully specified if quantification is given.
To conclude, we believe this research highlights how the SOM methodology provides a better solution for the liability assignment process with respect to the traditional one carried out in recent years. Therefore, this implies an important contribution to quasi-induced exposure method, which is used to estimate relative exposure, essential for road collision rate estimation. The latter is, in turn, key when evaluating the impact of road safety measures.
We have clarified this in the new version of the manuscript, by adding the following paragraph to the Conclusions section (line 516):
“From the point of view of the added value of this paper, this research includes important quantitative results. The paper thus goes clearly far beyond a purely binary or qualitative statement on whether given variables are influential or not on liability. This quantification includes the numbers of drivers in each cluster/node as well as the SOM topology and neighbor structure. A complete description of the patterns identified necessarily includes this information. Moreover, the borders between clear, non-clear and intermediate liability assignments are also only fully specified if quantification is given.”
Reviewer 3 Report
I am unable to comment on the actual statistical methods used here. Assuming the methods are sound, this is one the best-written papers I have read in recent years. The conclusions seem sound too, and I believe the findings are really useful in furthering the narrative on what is important for road safety.
Author Response
Thank you very much for your kind comment.
Reviewer 4 Report
The manuscript aims to explore driver liability in traffic crashes in Spain.
The topic of the study is relevant to journal and important.
Although the topic is crucial, it has not been comprehensively investigated.
The visual clustering technique, self-organizing maps method was applied and found that alcohol/drug use could be influential on liability
and further analysis is required for disability and sudden illness. The overall paper is well written and easy to follow. The conclusions drawn were supported by the data.
The manuscript aims to explore driver liability assessment in vehicle collisions using data of Spain. The topic is of importance and worthy of investigation. The reviewer enjoyed reading the manuscript. Just a small suggestion: need to add a paragraph to explain limitations of the study. Also, English can be improved.
Author Response
Thank you very much for your comments. The English has been revised and improved in all the manuscript. In addition, we have taken into account your suggestion (on the limitations of the study) to improve our paper.
We have clarified this in the new version of the manuscript, by adding the following paragraph to the “Conclusions” section (line 505):
“The main limitations of the study are: (a) it is very important to choose properly the SOM variables because their number is limited by its visibility (because it is a projected clustering), which is, as the same time, the main advantage of this clustering technique with respect to other ones. SOM thus provides a very quick assimilation/description of the cluster structure of the data. One may thus pay a small price for this advantage in terms of information; (b) quasi-induced exposure is a method based on driver behavior and condition. Therefore, to assign driver liability, traditionally only driver behavior and condition - related variables have been used. However, there are other variables related to vehicle collision circumstances, such as weather or pavement conditions, which could also affect, to a lesser degree, driver liability. Therefore, this issue should be explored in future research; and (c) as mentioned in the paper, more studies should be carried out with disability and sudden illness variables in order to deeply evaluate their relevance on driver liability.”
Reviewer 5 Report
The paper is well organized, and its purpose is clear.
The authors propose a method for analysing and assessing the driver liability in vehicle collisions.
Although the methodology is well presented and its application to the case study presents results in line with the scientific literature, it would be interesting to evaluate other types of variables included in the dataset that refer to the infrastructure conditions (e.g. pavement condition, radius of curve, etc.) and the environment conditions (e.g. weather, side obstacles, etc.).
Driver’s liability could be affected by parameters not explored in the study.
Author Response
Thanks for your significant comment. This paper proposes contributions to improve the liability assignment procedure traditionally applied in quasi-induced exposure method, which is based on the analysis of driver behavior and conditions, namely driver and speed offences. Here we propose using, as a starting point, 8 variables instead, i.e., more information.
This is why only driver behavior and condition -related variables have been used, but your suggestion of incorporating additional variables should be explored in future research. However, it is very important to choose properly the SOM variables because their number is limited by its visibility, which is, as the same time, the main advantage of this clustering technique with respect to other ones. SOM thus provides a very quick assimilation/description of the cluster structure of the data. One may thus pay a small price for this advantage in terms of information.
We have clarified this in the new version of the manuscript, by adding the following paragraph to the “Materials and Methods” section (“Database” subsection, line 180):
“In addition, other variables related to the collision characteristics or vehicle could be also considered. However, the quasi-induced exposure method is based on the driver behavior and conditions. Therefore, only driver behavior and condition - related variables are considered.”
And by adding the following paragraph to the “Conclusion” section (line 505):
“The main limitations of the study are: (a) it is very important to choose properly the SOM variables because their number is limited by its visibility, which is, as the same time, the main advantage of this clustering technique with respect to other ones. SOM thus provides a very quick assimilation/description of the cluster structure of the data. One may thus pay a (small) price for this advantage in terms of information; (b) quasi-induced exposure is a method based on driver behavior and condition. Therefore, to assign driver liability, traditionally only driver behavior and condition - related variables have been used. However, there are other variables related to vehicle collision circumstances, such as weather or pavement conditions, which could also affect, to a lesser degree, driver liability. Therefore, this issue should be explored in future research; and […].”
This manuscript is a resubmission of an earlier submission. The following is a list of the peer review reports and author responses from that submission.
Round 1
Reviewer 1 Report
This paper is generally well written though there needs to be much more explanation of terms used ( I have annotated a PDF for you). I'm not sure that the approach adds any value above multiple regression models which can simultaneously evaluate explanatory factors. I also think that given the variation in risk related to different population groups (i.e. age, gender, social class) that these are essential in order to understand liability. The modelling does not tell us about injury severity either. Another issue around modelling is that most modes aim to achieve to come to a parsimonious solution, having 25 clusters does not seem a helpful solution. The question remains is what do we do with these clusters and how do we target countermeasures from a public health perspective.

Author Response
Thank you very much for your very useful and interesting comments. We have reviewed them, answered to them and rewritten the paper accordingly. We hope the revised version meets your requirements.
This paper is generally well written though there needs to be much more explanation of terms used ( I have annotated a PDF for you).
Many thanks for your comments. We have taken them into account for a new version of the paper.
1) I'm not sure that the approach adds any value above multiple regression models which can simultaneously evaluate explanatory factors.
Thank you very much for your comment. It is true that, a priori, a supervised analysis by means of a classification model, either through traditional statistical methods such as logistic regression or machine learning techniques such as Random Forests, neural networks, support vector machines, boosting , etc. in which the output were liability, would be the ideal solution. The reason for not applying these techniques in this paper is the lack of labels (liable or not) in the accident data bases. This drives us to apply unsupervised analysis such as the SOM clustering, in which liability labels are assigned to clusters by expert criteria based on driver behavior patterns identified by SOM. To apply unsupervised techniques is an important contribution to the analysis of driver liability.
We updated the manuscript, clarifying this important issue in the “Introduction” section, by adding the following paragraph (line 88):
“Given that there are no liability labels in the vehicle collision records, it is not possible to apply supervised analysis techniques such as logistic regression or multiple regression models, to estimate driver liability in terms of driver offences and condition variables. Therefore, an unsupervised analysis, which takes into account these variables, is called for.”
2) I also think that given the variation in risk related to different population groups (i.e. age, gender, social class) that these are essential in order to understand liability. The modelling does not tell us about injury severity either.
Thanks for your significant comment. This paper proposes contributions to improve the liability assignment procedure traditionally applied in quasi-induce exposure method, which is based on the analysis of driver behavior and conditions. These “traditional” procedures usually only take into account driver and speed offences but here we propose using, as starting point, 8 variables instead, i.e., more information.
Moreover, the number of variables in a SOM is limited by its main advantage with respect to other clustering techniques: visibility (and hence quickness of understanding of the clustering structure). However, it is true that one may pay a price for this advantage in terms of information, because we are projecting onto a lower dimensional space. By using 8 variables, we consider that we have reached a satisfactory trade-off between clarity (visibility) and amount of information according to the literature reviewed.
This is why only -driver behavior and condition -related variables have been used.
We updated the manuscript to clarify this important issue in the “Database” subsection by adding the next paragraph (line 159):
“All driver behavior and condition - related variables have been considered, given that this research is a contribution to improve liability assignment. The latter needs to be applied for quasi-induced exposure, which is in turn driver behavior-based, as pointed out by other authors [6–8, 19]. As mentioned above, the “traditional” procedures of quasi-induced exposure method usually only take into account driver and speed offences, but here 8 variables are used instead, i.e., more information. In Table 1 the potentially relevant variables and their most common or representative types, are shown.”
In addition, we updated the manuscript to complete the “Methodology” subsection by adding the SOM drawbacks. In particular the one related to the number of variables (line 209):
“Moreover, the number of variables in a SOM is limited by its main advantage with respect to other clustering techniques: visibility (and hence quickness of understanding of the clustering structure). One may thus pay a price for this advantage in terms of information.”
3) Another issue around modelling is that most modes aim to achieve to come to a parsimonious solution, having 25 clusters does not seem a helpful solution. The question remains is what do we do with these clusters and how do we target countermeasures from a public health perspective.
Thanks for your significant comment. SOMs have been obtained by the authors with different map sizes, starting with the smallest ones, and 25 clusters have been chosen as the best trade-off between clarity and amount of information. In a smaller map, clusters could be too heterogeneous and larger maps would have too small cluster sample sizes. Moreover, 25 clusters can provide a better understanding of the multivariate structure of the data, beyond liability assignment. By means of the SOM methodology applied here, it is possible to create clusters in terms of driver offences. This allows for the identification of patterns which may result from a single cluster or a set of them. Subsequently, liability would be estimated by expert judgement in terms of the patterns. This pattern identification would be of use in design of countermeasures.
We have clarified this in the new version of the manuscript, by adding the following paragraph to the “Results and Discussion” section (“SOM division of the multivariate structure” subsection, line 313):
“There do exist quantitative criteria for selecting the number of clusters. For example, in probabilistic mixture model-based clustering, the EM algorithm may be applied. Here, the choice of 25 nodes was made empirically by trial and error: SOMs with different map sizes have been obtained, starting with the smallest ones. 25 nodes were considered a reasonable choice, given the trade-off sought between properly identifying patterns (clarity/visibility) and sample size per cluster: with an excessively small map size, clusters could be too heterogeneous and, therefore, adequate patterns would not be extracted; the same would occur with a very large map size, resulting in too small cluster sample sizes [39]. Also, 25 clusters is an adequate number to provide a better understanding of the multivariate structure of the data.”
Reviewer 2 Report
This manuscript aimed to identify accident liability patterns using Self-Organizing Maps (SOM). The results showed that alcohol and drug use should be considered as important factors that determine the liability. Such topic should be of interests to readers of IJERPH, however, the following issues should be considered before publication.
- I think the major issue is how to justify the accuracy of the classification method proposed in this study. Is it possible to get the liability information of these accidents from the police and compare your results with these “ground truth”?
- Line 62, the authors have reviewed studies investigating factors related to collision liability. However, there should be rules and laws to be followed by the polices or judges to determine liability in an accident, rather than just looking at the hazardous driving behavior.
- Line 117, it was said that “the standardization of both databases has not been completed”. I wonder what kind of standardization it is and if the incompleteness would affect your results.
- Another problem is that many of the statements/methodologies in the paper lack scientific support. For instance, Table 1 lists the variables considered. Why these variables were chosen, any references to support Table 1? Line 136, why the interactions between “the rest of the variables” would determine liability, given that “they alone are believed to be non-determinant”? The authors need to provide the rationale for making the above assumption. Similarly, line 338, the authors established a boundary to separate liable and presumed non-liable drivers, based on “expert criteria”. While “expert criteria” can be used as scientific evidence or support, the authors need to specify what the criteria were.
Author Response
Thank you very much for your very useful and interesting comments. We have reviewed them, answered to them and rewritten the paper accordingly. We hope the revised version meets your requirements.
This manuscript aimed to identify accident liability patterns using Self-Organizing Maps (SOM). The results showed that alcohol and drug use should be considered as important factors that determine the liability. Such topic should be of interests to readers of IJERPH, however, the following issues should be considered before publication.
- I think the major issue is how to justify the accuracy of the classification method proposed in this study. Is it possible to get the liability information of these accidents from the police and compare your results with these “ground truth”?
Thanks for your significant comment.
The Spanish vehicle collision databases do not provide a liability assessment by police officials. If available, this information would be of some use, not only as a “test set” to validate the results of the liability estimation/prediction, but also, previously, to build a supervised learning model in which liability were the response. However, there exists, in police citations, the risk of “negative halo bias” (De Young et al., 1997). An example of this phenomenon was illustrated by Jiang et al. (2012), who discovered that young male drivers using alcohol/drugs were more likely to be called for deposition by the police, which would in turn bias police citation sampling (Jiang and Lyles, 2010; Jiang and Lyles, 2011; Jiang et al., 2012; Jiang, Lyles and Guo, 2014).
This is why, in the absence of liability labels, a powerful and visual unsupervised learning technique, called Self-Organizing Map (SOM), has been applied. This method can help to assign driver liability labels by expert judgment.
We have clarified this in the new version of the manuscript, by rewriting the following paragraph to the “Introduction” section (line 72):
“However, adding information on non-driving behavior or on the driver’s state citations could result in uncertainty or statistical bias in exposure estimation [5–8, 20, 26]. For example, there exists in police citations, the risk of “negative halo bias” [26]. An example of this phenomenon was illustrated by [7], who discovered that young male drivers using alcohol/drugs were more likely to be called for police citation, which would in turn bias citation sampling [6-8].”
- Line 62, the authors have reviewed studies investigating factors related to collision liability. However, there should be rules and laws to be followed by the polices or judges to determine liability in an accident, rather than just looking at the hazardous driving behavior.
Thanks for your significant comment. This paper proposes contributions to improve the liability assignment procedure traditionally applied in quasi-induce exposure methods, which are based on the analysis of driver behavior and conditions. These “traditional” procedures usually only take into account driver and speed offences but here we propose using, as starting point, 8 variables instead, i.e., more information. This is why only -driver behavior and condition -related variables have been used.
Moreover, with the exception of very clear situations, the authors are not aware of the existence of a set of clear rules for liability assignments by police officials or judges, which in any case, would have been provided by road safety experts. Moreover, in Spain, only fatal accidents are taken to the courtrooms.
We have clarified this in the new version of the manuscript, by adding the following paragraph to the “Materials and methods” section (“Database” subsection, line 159):
“All driver behavior and condition - related variables have been considered, given that this research is a contribution to improve liability assignment. The latter needs to be applied for quasi-induced exposure, which is in turn driver behavior-based, as pointed out by other authors [6–8, 19]. As mentioned above, the “traditional” procedures of quasi-induced exposure method usually only take into account driver and speed offences, but here 8 variables are used instead, i.e., more information.”
- Line 117, it was said that “the standardization of both databases has not been completed”. I wonder what kind of standardization it is and if the incompleteness would affect your results.
Thanks for your significant comment. The term “standardization” could be confusing, we mean that the two databases are not built with exactly the same structure: there is lack of homogeneity, mainly for fatalities, since follow-up is recorded mainly in hospitals. Additionally, for the ARENA database there is a new coding for vehicles. Work is currently being carried out so that, in future research, information on the recent years may be added, thus increasing the size and hence also the accuracy of the statistical inference. However, a large sample size is used for the research in this paper.
We have clarified this in the new version of the manuscript, by adding the following paragraph to the “Materials and methods” section (“Database” subsection, line 142):
“[…]and so far, the two databases have different format and structure, mainly for fatalities. Work is currently being carried out so that, in future research, information on the recent years may be added, thus increasing the size and hence also the accuracy of the statistical inference. However, a large sample size is used for the research in this paper.”
- Another problem is that many of the statements/methodologies in the paper lack scientific support. For instance, Table 1 lists the variables considered. Why these variables were chosen, any references to support Table 1? Line 136, why the interactions between “the rest of the variables” would determine liability, given that “they alone are believed to be non-determinant”? The authors need to provide the rationale for making the above assumption. Similarly, line 338, the authors established a boundary to separate liable and presumed non-liable drivers, based on “expert criteria”. While “expert criteria” can be used as scientific evidence or support, the authors need to specify what the criteria were.
Thanks for your significant comment.
This paper proposes contributions to improve the liability assignment procedure traditionally applied in quasi-induce exposure method, which is based on the analysis of driver behavior and conditions. These “traditional” procedures usually only take into account driver and speed offences but here we propose using, as starting point, 8 potentially relevant (driver behavior and condition) variables instead, i.e., more information. Moreover, the number of variables in a SOM is limited by its main advantage with respect to other clustering techniques: visibility (and hence quickness of understanding of the clustering structure). However, it is true that one may pay a price for this advantage in terms of information. Because we are projecting onto a lower dimensional space. By using 8 variables, we consider that we have reached a satisfactory trade-off between clarity (visibility) and amount of information according to the literature reviewed.
This is why only -driver behavior and condition -related variables have been used.
On the comment about line 136, the joint effect of subset of variables could be more relevant in liability than their individual effects. Therefore, there are driver behavior patters regarding liability that would only come to light when several variables are studied together.
Finally, on the expert criteria, a boundary has been estimated in the map, depending on the “subset of offences committed by drivers” pattern in each cluster. Given that SOM is an unsupervised analysis tool, and thus no labels exist in the data, there is some unavoidable degree of subjectivity inherent to labelling. The upper region of the map, as established by the boundaries, includes all drivers who have committed offences which, in accordance with the literature reviewed, carry very large weight when establishing liability, e.g., speed offences. However, in the lower region, we find those clusters which include all drivers whose offences, as viewed traditionally in the literature, would not determine liability per se.
We consider that adding the last paragraph we were already including when answering your second comment meets the first part of this last (fourth) one.
The second part of the fourth comment has been clarified in the new version of the manuscript, by adding the following paragraphs to a) the “Materials and methods” section and b) to the “Patterns identification for the liability assignment process” subsection:
In the “Materials and methods” section (“Database” subsection, line 172):
“[…]it is possible that the interactions between several of them will be influential because the joint effect of subset of variables could be more relevant in liability than their individual effects.”
In “Pattern identification for the liability assignment process” subsection (line 400):
“[…] i.e., the boundary in the map has been estimated depending on the “subset of offences committed by drivers” pattern in each cluster.
At the top of the map (68,730 at-fault drivers), the labeling errors should be minimal, given their profile of offences because this area includes all drivers who have committed offences which, in accordance to the literature reviewed, carry very large weights when establishing liability, e.g. speed offences.”
Reviewer 3 Report
This paper uses the visual clustering technique Self-Organizing Maps (SOM) to analysis on the multivariate structure in the data and select the most important variables on driver liability. This is meaningful work, but there are some problems that need to be improved.
(1)Introduction: The author must describe the contribution of the paper in more detail. Let the reader understand the academic significance of the article.
(2)Materials and Methods:Please provide more details on how data from multiple drivers involved in a traffic accident is used. Is it one data point or two data points? Is the number of drivers mentioned in the article equal to the number of accidents?
The SOM model is selected in this article. Please explain in detail why?
Author Response
Dear editor,
Thank you very much for your very useful and interesting comments. We have reviewed them, answered to them and rewritten the paper accordingly. We hope the revised version meets your requirements.
This paper uses the visual clustering technique Self-Organizing Maps (SOM) to analysis on the multivariate structure in the data and select the most important variables on driver liability. This is meaningful work, but there are some problems that need to be improved.
(1) Introduction: The author must describe the contribution of the paper in more detail. Let the reader understand the academic significance of the article.
Thank you for pointing out this important issue. The main contributions of this paper are:
- To explore the inclusion of more variables in the liability assignment procedure, given that, in the literature, only driver and speed offences are taken into account to assign driver liability.
- To apply a powerful clustering (thus unsupervised analysis) technique such as SOM, for expert-judgement based liability assignment in terms of the patterns identified from the clustering. Since no liability labels exist in the data bases, it is necessary to apply unsupervised analysis techniques, which would allow to indirectly extract information on liability, in terms of the variables which are available in the data bases. Therefore, we carry out SOM cluster analysis with this larger set of variables. Using this clustering and expert judgment, liability regions are identified in terms of (sometimes complex) multivariate patterns. In addition, as an added value, the SOM provides a better understanding of the multivariate structure of the data. The use of an unsupervised analysis technique, such as SOM, implies an important methodological contribution to liability assignment, given that to date there exists no systematic statistical methodology to this end, and that there are potentially relevant variables which were not taken into account in the literature and could also affect liability.
We have clarified this in the new version of the manuscript, by adding the following paragraph to the “Introduction” section (line 99):
“Therefore, the main contributions of this paper are: (1) To explore the inclusion of more variables in the liability assignment procedure, given that in the literature, only driver and speed offences are taken into account to assign driver liability and (2) To apply a powerful clustering (thus unsupervised analysis) technique such as Self-Organizing Maps (SOM), for expert judgment-based liability assignment in terms of the patterns identified from the clustering. Using this clustering and expert judgment, liability regions are identified in terms of (sometimes complex) multivariate patterns. In addition, as an added value, the SOM also provides a better understanding of the multivariate structure of the data. The use of an unsupervised analysis technique, such as SOM, implies an important methodological contribution to liability assignment, given that to date there exists no systematic statistical methodology to this end, and that there are potentially relevant variables which were not taken into account in the literature and could also affect liability.”
(2)Materials and Methods:Please provide more details on how data from multiple drivers involved in a traffic accident is used. Is it one data point or two data points? Is the number of drivers mentioned in the article equal to the number of accidents?
Thank you for your significant comment.
The data base used to carry out this research consists of records with information on drivers involved in crashes between two passenger cars. Thus, the number of drivers analyzed is equal to the number of records and twice the number of vehicle collisions. There were three reasons for the choice of passenger cars. First, according to the data from the Spanish General Traffic Directorate, passenger cars account for more than 70% of vehicles. Secondly, the number of victims in interurban road crashes is also above 70% of the grand total. Third, passenger cars are the group where quasi-induced exposure methods have been more frequently applied. It is thus a very important group for road safety research.
In addition, interurban areas were chosen because the number of killed and seriously injured drivers in interurban areas is much larger than in urban ones. In particular, in the 2004-2013 period analyzed here, the figures for killed and seriously for interurban areas were 4 times and twice those of the urban ones, respectively
We have clarified this in the new version of the manuscript, by adding the following paragraph to the “Database” subsection (line 127):
“[…] Thus, the number of drivers analyzed is equal to the number of records and twice the number of vehicle collisions.
There were three reasons for the choice of passenger cars. First, according to the data from the Spanish General Traffic Directorate, in the study period of this paper (2004-2013), passenger cars account for more than 70% of vehicles. Secondly, the number of victims in interurban road crashes is also above 70% of the grand total. Third, passenger cars are the vehicle group where quasi-induced exposure methods have been more frequently applied. It is thus a very important group for road safety research.
In addition, interurban areas were chosen because the number of killed and seriously injured drivers in interurban areas is much larger than in urban ones. In particular, in the 2004-2013 period analyzed here, the figures for killed and seriously for interurban areas were 4 times and twice those of the urban ones, respectively.”
3) The SOM model is selected in this article. Please explain in detail why?
Thank you very much for your significant comment. To start with, the absence of liability labels in the data calls for the use of unsupervised learning techniques, such as cluster analysis. Within clustering tools, SOM has, with respect to other clustering techniques, the advantage of visibility -resulting from projecting onto a smaller (2 or 3D) and thus “visible” dimension-and quicker assessment of the clustering structure.
We have clarified this in the new version of the manuscript by adding a new paragraph in “Methodology” subsection (line 213).
“The choice of SOM is justified because the absence of liability labels in the data calls for the use of unsupervised learning techniques, such as cluster analysis. Within clustering tools, SOM has, with respect to other clustering techniques, the advantage of visibility […]”
Reviewer 4 Report
This is a very interesting paper describing a methodology for assigning drivers liability to crashes where two passenger’s cars are involved. All the paper’s sections are well written and it could be recommended for publication after some minor changes.
The authors should consider the following remarks and suggestion for further improvement of their paper:
*The authors should better explain why they do not consider other types of vehicles (trucks, motorcycles, etc.) and they limit their analysis to accidents with two passenger cars.
*There are many data mining criteria for selecting the number of clusters. The authors should better justify why they use “expert judgment” as criteria.
*Table 3 should also include a column with the number of cases per cluster. How do you explain that there are two clusters (5 and 15) that represent more than 50% of the sample?
*The authors should clarify the process for categorizing the influential variables as shown in table 4. It seems that this categorization is mainly qualitative or based on expert judgment, and it is not very clear that it proceeds from the cluster analysis.
*The authors should acknowledge the main limitations of the SOM methodology.
*Tables 5 and 6 compare the results using the SOM methodology and the standard criteria (based on driver and speed offences only). As the paper has identify Alcohol/Drug use as one of the most influential variables, I would suggest to also include a table (new table 7) with the same classification as in tables 5/6 in order to compare the SOM results with a method based on the three most influential variables. From a practical point of view, I think that SOM methodology will be difficult to apply for practitioners (or even other researches), but a recommendation for adding a new variable (Alcohol/Drug use) to the standard procedure could be an interesting proposal for future studies.
Please, in order to improve your manuscript, include your comments to the previous point in the MAIN TEXT, not only in the response to reviewers.
Author Response
Dear editor,
Thank you very much for your very useful and interesting comments. We have reviewed them, answered to them and rewritten the paper accordingly. We hope the revised version meets your requirements.
This is a very interesting paper describing a methodology for assigning drivers liability to crashes where two passenger’s cars are involved. All the paper’s sections are well written and it could be recommended for publication after some minor changes.
The authors should consider the following remarks and suggestion for further improvement of their paper:
Please, in order to improve your manuscript, include your comments to the previous point in the MAIN TEXT, not only in the response to reviewers.
Thank you very much for your comments.
- *The authors should better explain why they do not consider other types of vehicles (trucks, motorcycles, etc.) and they limit their analysis to accidents with two passenger cars.
Thank you very much for your significant comment. In accordance with the data from the Spanish General Traffic Directory, in the study period of this paper (2004-2013), passenger cars account for more than 70% of all vehicles. Additionally, the number of victims in interurban road crashes along this period was also above 70% of the grand total. Moreover, passenger cars are the vehicle group to which quasi-induced exposure method are most frequently applied. This is why passenger cars were selected for our application, but the methodology could be applied to other vehicle types.
We have clarified this in the new version of the manuscript, by adding the following paragraph to the “Database” subsection (line 129):
“There were three reasons for the choice of passenger cars. First, according to the data from the Spanish General Traffic Directorate, in the study period of this paper (2004-2013), passenger cars account for more than 70% of vehicles. Secondly, the number of victims in interurban road crashes is also above 70% of the grand total. Third, passenger cars are the vehicle group where quasi-induced exposure methods have been more frequently applied. It is thus a very important group for road safety research.”
- *There are many data mining criteria for selecting the number of clusters. The authors should better justify why they use “expert judgment” as criteria.
Thank you very much for your significant comment. There do exist quantitative criteria for selecting the number of clusters. For example, in probabilistic mixture model-based clustering, the EM algorithm may be applied. Here we used an empirical trial and error method, trying to reach the trade-off sought between properly identifying patterns (clarity/visibility) and sample size per cluster. In a smaller map, clusters could be too heterogeneous and larger maps would have too small cluster sample sizes.
We have clarified this in the new version of the manuscript, by adding the following paragraph to the “Results and Discussion” section (“SOM division of the multivariate structure” subsection, line 313):
“There do exist quantitative criteria for selecting the number of clusters. For example, in probabilistic mixture model-based clustering, the EM algorithm may be applied. Here, the choice of 25 nodes was made empirically by trial and error: SOMs with different map sizes have been obtained, starting with the smallest ones. 25 nodes were considered a reasonable choice, given the trade-off sought between properly identifying patterns (clarity/visibility) and sample size per cluster: with an excessively small map size, clusters could be too heterogeneous and, therefore, adequate patterns would not be extracted; the same would occur with a very large map size, resulting in too small cluster sample sizes [39]. Also, 25 clusters is an adequate number to provide a better understanding of the multivariate structure of the data.”
- *Table 3 should also include a column with the number of cases per cluster. How do you explain that there are two clusters (5 and 15) that represent more than 50% of the sample?
Thank you very much for your comment.
Table 3 has been updated to include the number (and percentage) of drivers in each cluster (line 336).
Cluster 5 is formed by all drivers who committed no offences and presented no unfavorable conditions for driving, this is thus why it includes so many drivers. On the other hand, cluster 15 includes all drivers who have committed some driver offence. There are 22 types of such offences and, additionally, some of them are relatively frequent (e.g. distractions). Thus, this cluster also contains many drivers.
We have clarified this in the new version of the manuscript, by adding the following paragraph to the “SOM division of the multivariate structure” subsection (line 341):
“In accordance with the information presented in figure 5 and table 3, there are two important SOM clusters, number 5 and number 15, which include more than 50% of the total driver sample. Cluster 5 is formed by all drivers who committed no offences and presented no unfavorable conditions for driving, this is thus why it includes so many drivers. On the other hand, cluster 15 includes all drivers who have committed some driver offence. There are 22 types of such offences and, additionally, some of them are relatively frequent (e.g. distractions). Thus, this cluster also contains many drivers.”
- *The authors should clarify the process for categorizing the influential variables as shown in table 4. It seems that this categorization is mainly qualitative or based on expert judgment, and it is not very clear that it proceeds from the cluster analysis.
Thank you very much for your significant comment.
In order to categorize the variables, in the first place a numeric value (2) is assigned to all cases where the offence of negative condition do exist and another (0) for those where they do not. The range between clear offences and clear absence of them is thus 2 for all offence-related variables. Their ranges are then “standardized”. A problem appears, however, when defining a value for cases where the variable is unknown. To this end, two hypotheses (see page 6, Categorization of the variables section of paper) are used a starting point to decide on which values should be assigned in such cases. SOM is also used here by means of a SOM-based sensitivity analysis carried out to assess the effect of the hypotheses above on the results.
We have clarified this in the new version of the manuscript, by adding the following sentence to the “Categorization of the variables” section (line 237):
“Thus, to assess this question from the point of view of its effect on the SOM, in this section, a SOM-based sensitivity analysis using a sample of the database is carried out […]”
- *The authors should acknowledge the main limitations of the SOM methodology.
Thanks for your significant comment.
A certain limitation of SOM is that, since it performs multicriteria optimization (on both cluster homogeneity and conservation of topology), then if one just wanted to optimize on the first criterion, it (SOM) would be suboptimal and one should use traditional (non-projected) cluster techniques (e.g. K-means) instead.
Moreover, the number of variables in a SOM is limited by its main advantage with respect to other clustering techniques: visibility (and hence quickness of understanding of the clustering structure). One may pay a price for this advantage in terms of information. By using eight variables, we consider that we have reached a satisfactory trade-off between clarity (visibility) and amount of information according to the literature reviewed.
We have clarified this in the new version of the manuscript, by adding the following paragraph to the “Methodology” subsection (line 205):
“A certain limitation of SOM is that, since it performs multicriteria optimization (on both cluster homogeneity and conservation of topology), then if one just wanted to optimize just on the first criterion, it (SOM) would be suboptimal and one should use traditional (non-projected) cluster techniques (e.g. K-means) instead.
Moreover, the number of variables in a SOM is limited by its main advantage with respect to other clustering techniques: visibility (and hence quickness of understanding of the clustering structure). One may thus pay a price for this advantage in terms of information.”
- *Tables 5 and 6 compare the results using the SOM methodology and the standard criteria (based on driver and speed offences only). As the paper has identify Alcohol/Drug use as one of the most influential variables, I would suggest to also include a table (new table 7) with the same classification as in tables 5/6 in order to compare the SOM results with a method based on the three most influential variables. From a practical point of view, I think that SOM methodology will be difficult to apply for practitioners (or even other researches), but a recommendation for adding a new variable (Alcohol/Drug use) to the standard procedure could be an interesting proposal for future studies.
Thank you very much for your significant comment. We consider this comment very interesting.
The patterns and thus the boundaries identified by SOM are more complex than those which would result from just taking into account driver and speed offences and alcohol/drugs. Some of the remaining variables may not be very relevant marginally, but could be so jointly with others. Thus, although these three variables are the most relevant, significant information is still contained in the remaining ones and there would be significant differences between the classifications obtained from the two approaches. Moreover, although SOM is, as the reviewer indicates, a sophisticated technique, the use of the ad-hoc R software facilitates enormously its application. We would then encourage researchers to apply it anyway. Additionally, the hierarchy of variable importance and patterns extracted for interurban could change when applying the methodology to urban ones.
We have clarified this in the new version of the manuscript, by adding the following paragraph to the “Conclusion” section (line 483):
“The analysis suggests that the most important variables for driver liability are: Driver offence, speed offence and alcohol/drug use, although the two first ones are the most important. The relevance of the variables disability and sudden illness regarding the driver liability is not clear. Therefore, more studies should be carried out with these two variables. Administrative offence, drowsiness and vehicle defect are clearly the less relevant ones, in such way that, when carrying out the liability assignment, not taking them into account would not be influential, thus simplifying the process (with respect to considering the eight initial offence variables). However, the patterns and thus the boundaries identified by SOM are more complex than those which would result from just taking into account driver and speed offences and alcohol/drug use. Some of the remaining variables may not be very relevant marginally, but could be so jointly with others. Thus, although the three first variables are the most relevant, some information is still contained in the two following ones.”
Round 2
Reviewer 1 Report
Thank you to the authors for their response. However, the lack of data on basic demographics and the known risk profiles associated them makes this approach to analysis of road casualties too global and of little utility to policy makers.
Reviewer 2 Report
Thanks for addressing all my comments. I have no further questions.